# Sperm Proteomics Analysis of Diabetic Induced Male Rats as Influenced by *Ficus carica* Leaf Extract

**Umarqayum Abu Bakar [1] , Puvaratnesh Subramaniam [1], Nurul Ain Kamar Bashah [1],**
**Amira Kamalrudin [1], Khaidatul Akmar Kamaruzaman [1], Malina Jasamai [2] ,**
**Wan Mohd Aizat [3] , M. Shahinuzzaman [4] and Mahanem Mat Noor [1,\***

[1] Centre for Biotechnology and Functional Food, Faculty of Science and Technology, Universiti Kebangsaan Malaysia, Bangi 43600, Malaysia; umarqayum92@gmail.com (U.A.B.); puvaratnesh@hotmail.com (P.S.); nurulain_0917@yahoo.com (N.A.K.B.); amirakamalrudin@gmail.com (A.K.); khaidatulakmar89@gmail.com (K.A.K.)

[2] Faculty of Pharmacy, Universiti Kebangsaan Malaysia, Kuala Lumpur 50300, Malaysia; malina@ukm.edu.my

[3] Institute of Systems Biology (INBIOSIS), Universiti Kebangsaan Malaysia, Bangi 43600, Malaysia; wma@ukm.edu.my

[4] School of Chemical Sciences and Food Technology, Faculty of Science and Technology, Universiti Kebangsaan Malaysia, Bangi 43600, Malaysia; shahinchmiu@gmail.com

\* Correspondence: mahanem@ukm.edu.my; Tel.: +603-8921-5193

**Abstract:** Diabetes mellitus is shown to bring negative effects on male reproductive health due to long-term effects of insulin deficiency or resistance and increased oxidative stress. *Ficus carica* (FC), an herbal plant, known to have high antioxidant activity and antidiabetic properties, has been used traditionally to treat diabetes. The objective of this study is to determine the potential of the FC leaf extract in improving sperm quality of streptozotocin (STZ) induced diabetic male rats from proteomics perspective. A total of 20 male rats were divided into four groups; normal (nondiabetic rats), negative control (diabetic rats without treatment), positive control (diabetic rats treated with 300 mg/kg metformin), and FC group (diabetic rats treated with 400 mg/kg FC extract). The treatments were given via oral gavage for 21 consecutive days. The fasting blood glucose (FBG) level of FC treated group demonstrated a significant ($p < 0.05$) decrease compared to negative group after 21 days of treatment, as well as a significant ($p < 0.05$) increase in the sperm quality parameters compared to negative group. Sperm proteomics analysis on FC treated group also exhibited the increase of total protein expression especially the proteins related to fertility compared to negative group. In conclusion, this study clearly justified that FC extract has good potential as antihyperglycemic and profertility agent that may be beneficial for male diabetic patients who have fertility problems.

**Keywords:** *Ficus carica*; diabetes mellitus; proteomics; sperm quality

## 1. Introduction

According to the World Health Organization [1], there were 422 million people with diabetes worldwide in 2014. Diabetes causes metabolic disturbances of carbohydrate, fat, and protein that result in complications of the organ and body systems such as retinopathy, nephropathy, and neuropathy diabetic [2]. Diabetes also affects the function of the male reproductive system by decreasing sperm concentration, viability, and increasing sperm apoptosis [3,4]. The male reproductive system is affected by diabetes probably due to the insulin deficiency or resistance, and the increase of oxidative stress. Insulin deficiency or resistance is found to directly affect spermatogenesis, testis development, and the secretion of hormone related to male fertility due to the failure of glucose transportation into sertoli

cells [5]. Meanwhile, Ding et al. [6] reported that the increase of oxidative stress due to hyperglycemia causes the excessive production of free radicals that eventually induce sperm DNA fragmentation, and reduce the expression of fertility protein.

The emergence of proteomic approach helps researchers discover precisely the root cause of male infertility due to diabetes. Proteomics is the identification and quantification of proteins that give an understanding about protein function and its interactions with other proteins, hence lead to the understanding of certain expressed phenotype [7,8]. Based on a compilation by Amaral et al. [9], there are 6198 different proteins in human spermatozoa, and each protein has its own distinctive role either directly or indirectly in male fertility. In general, diabetes alters the expression of these sperm fertility proteins that finally results in male infertility. An et al. [10] reported that diabetes reduces sperm proteins such as cystatin C and dipeptidyl peptidase 4 which play vital roles in mitochondrial metabolism that affect directly the motility of sperm. While Pavlinkova et al. [4] found that diabetes changes protamine 1/protamine 2 ratio that indicates reduced sperm quality, since protamines are small proteins that bind sperm DNA which are important for DNA stability and sperm maturation. Accurate representation of male infertility due to diabetes by proteomics analysis of sperm protein eases the determination of effective treatment [10].

Currently, diabetes is treated by a synthetic drug such as metformin to prevent diabetes complications, but the treatment does not significantly improve sperm quality parameters [11,12] and somehow it reduces sperm quality parameters [13–15]. As an option, the focus has now shifted to the use of herbs to serve as therapeutic alternatives. *Ficus carica* (FC), also known as the fig, is among the potent medicinal plants in treating diabetes. Irudayaraj et al. [16] suggest that FC is able to enhance the insulin sensitivity in diabetic rats via the repair of one or more defects such as insulin receptor, insulin receptor substrate, or glucose transporter proteins, as demonstrated by the decrease of high blood glucose level, normalization of plasma insulin, and improvement of insulin tolerance level in FC treated group. A previous study also reported the profertility effect of FC on infertile male animal models. Naghdi et al. [17] found that FC improves testis tissue condition, as well as increases sperm number and progressive sperm percentage of formaldehyde-induced mice. FC leaf extract is effective in managing diabetes and also treating male infertility, however, the proteomic data related to its profertility potential on diabetic subjects is not available.

Therefore, this study aims to explore the profertility effect of FC leaf extract on diabetic-induced male rats at particular genetic sequences in protein level.

## 2. Materials and Methods

### 2.1. Preparation of FC Aqueous Extract

FC leaves (cultivar B110) were collected from Saf Fa Fig Garden at Kuala Pilah, Negeri Sembilan, Malaysia, under the Department of Chemical and Process Engineering, Faculty of Engineering and Built Environment. FC leaves were deposited in the Universiti Kebangsaan Malaysia herbarium with the voucher number, UKMB40389. FC aqueous extract was prepared as described by Perez et al. [18]. Briefly, FC leaves were rinsed with water, dried, and ground to fine powder. The fine powder was mixed with water in the ratio of 1:9, and boiled at 100 °C for 30 min. The extract was later filtered and freeze dried.

### 2.2. Experimental Animal

A total of 20 male Sprague-Dawley rats aged eight weeks were divided into four groups. Three groups were normal, positive (diabetic group treated with metformin at 300 mg/kg per body weight) and negative control (diabetic group without treatment), and one group was treated with FC aqueous extract at 400 mg/kg per body weight. All groups were treated by oral gavage every day for 21 consecutive days. The study was approved by the Animal Ethics Committee of Faculty of Medicine, Universiti Kebangsaan Malaysia (FST/2017/MAHANEM/29-MARCH/833-MARCH-2017-FEB.-2019).

### 2.3. Antihyperglycemic Activity

Diabetes was induced by a single intravenous injection of STZ (50 mg/kg) (Sigma – Aldrich, Saint Louis, Missouri, USA) dissolved in citrate buffer (0.1 M, pH 4.5) (Sigma – Aldrich, Saint Louis, Missouri, USA) after 16 h of fasting. Diabetes was confirmed by measuring fasting blood glucose (FBG) level from the tail tip after five days of STZ-induction with glucometer AccuCheck®Performa (Roche Diagnostic GmbH, Manheim, Germany). The FBG level of 13 mmol/L and above was considered as diabetic. The FBG level after treatment was measured on day 22.

### 2.4. Sperm Quality Analysis

All rats were sacrificed on day 22, and cauda epididymis was isolated, minced, and suspended in 15 mL of a Biggers-Whitten-Whittingham (BWW) medium. BWW is a medium used effectively to support the fertilizing potential of spermatozoa in vitro, supplemented with balanced salt, bovine serum albumin (BSA), energy substrates such as glucose, pyruvate, and lactate, as well as $Ca^{2+}$ and $HCO_3$ (Sigma – Aldrich, Saint Louis, Missouri, USA) [19]. The sperm preparation was then incubated in 5% of $CO_2$ incubator for 30 min at 37 °C to allow sperms to swim up. Sperm count, motility, and viability were assessed using an improved Neubauer haemocytometer in accordance to WHO 2010 laboratory manual. Meanwhile, sperm morphology was determined as described by Seed et al. [20].

Sperm count was assessed using a haemocytometer based on WHO protocol [21]. The sperm sample was pipetted on the middle grid of the haemocytometer, and counted at 10 boxes of that particular grid under light microscope with 100× magnification. Sperm count was recorded in millions ($10^6$). Then, the same haemocytometer was used in sperm motility determination. Sperm motility was presented in percentage values in accordance to the grade determined by WHO [21]; progressive (P), nonprogressive (NP), and immotile (IM). Next, based on the motility of the sperm, sperm viability was assessed by using the following formula:

$$\frac{\text{Viable sperm (P and NP)}}{200 \text{ sperm}} \times 100\%$$

Lastly, for normal sperm morphology analysis, the sperm sample was pipetted on the glass slide, and smeared over the slide surface. The smeared slide was then dried and stained using Giemsa staining before being observed under a light microscope.

### 2.5. Sperm Proteomic Analysis

A further study using shotgun proteomics approach was performed to identify and characterize the diabetic rat sperm protein profile in FC group in comparison with the control group. The sperm protein of control groups (normal, negative, and positive) and FC treatment group (dose 400 mg/kg) was used in the proteomic analysis. The protein extraction was conducted based on protocols by Yunianto et al. [22]. The sperm was harvested from the caudal epididymis before being suspended in a Biggers-Whitten-Whittingham (BWW) medium for 30 min at 37 °C in 5% of $CO_2$ incubator. The sperm samples were centrifuged and lysed with a lysis buffer. The major components of the lysis buffer were urea, 3-cholamidopropyl dimethylammonio 1-propanesulfonate (CHAPS), immobilized pH gradient (IPG) buffer, and phenylmethylsulfonyl fluoride (PMSF). Determination of sperm protein concentration was performed using a Bradford [23] assay to ensure that the extracted sample had sufficient concentration for gel electrophoresis and LCMS/MS analysis. The gel electrophoresis of protein sample was then performed at the voltage of 75 V using sodium dodecyl sulphate (SDS) gel 12.5% and broad range prestained protein markers (Nacalai Tesque, Japan). The protein band in gel was cut and incubated with dithiotreitol and iodoacetamide for reduction and alkylation steps, respectively. This is followed by an overnight incubation with 6 ng/μL trypsin (Promega, Madison, Wisconsin, USA) for protein digestion [24].

Mass spectrometry (MS) analysis was performed using nanoflow reversed phase liquid chromatography (Dionex 3000 Ultimate RSLCnano, Thermo Fisher Scientific, Waltham, MA, USA) coupled to the Orbitrap Fusion mass spectrometer (Thermo Scientific Orbitrap Fusion). The nanoLC system used was the EASY-Spray column Acclaim PepMap C18 100 A°, 50 μm id × 15 cm with particle size of 2 μm. Five microliters digested samples were injected and run in the chromatography system using the gradient mobile phase method with a constant flow of 250 nL/min. This consists of solvent A (0.1% formic acid in water) and solvent B (0.1% formic acid in acetonitrile) with linear gradient running conditions (91 min at 5%–40% B, 2 min at 85% B, 3 min at 85% B, 1 min at 5% B, and 4 min at 5% B). All chemicals used were LC-MS grade purchased from Fisher Scientific (Fair Lawn, NJ, USA).

The MS spectra was acquired with a scan range of 350–1800 m/z, 50 ms injection period, 120,000 resolution, and an accumulation gain control (AGC) target of 4.0 $e^5$ (400,000). Peptide precursors were selected for MS/MS based on a charge state of 2–7, an assigned monoisotopic m/z value, intensity threshold of 5000 and 20-s dynamic exclusion window. Selected precursors were fragmented using high-energy collision induced dissociation (HCD) (Thermo Fisher Scientific, Waltham, MA, USA) at 28% normalized collision energy. Ion trap MS (ITMS) (Thermo Fisher Scientific, Waltham, MA, USA) was used to analyze the MS/MS spectra with 1.6 m/z isolation window, 250 ms injection time, 60,000 resolving power, rapid scan rate, and $1.0e^2$ (100) AGC target.

Mass spectra data acquired was analyzed by Thermo Scientific Proteome Discoverer Software Version 2.1. The sequence database for rat (*Rattus norvegicus*) was obtained from UniProt database (http://www.uniprot.org) accessed on May 2018. The parameter search included tryptic specific digest with two or less miscleavages and residue modification was set as fixed (cysteine carbamidomethylation) and variable modifications (methionine oxidation and deamidation of aspargine and glutamine). During the main search, parent and fragment ions were permitted with a mass deviation of 10 ppm and 0.6 Da, respectively.

All identified proteins were annotated using Blast2GO software version 5.2 (https://www.blast2go.com) (BioBam Bioinformatics, Valencia, Spain). The blast parameter was set as blastp with an expectation value (E-value) $1 \times 10^{-3}$ against UniProt database. Then, protein sequences were further examined using InterproScan, Mapping, and Annotation. The gene ontology graph was generated using WEGO (Web Gene Ontology Annotation Plot) program version 2.0 (http://wego.genomics.org.cn) (WEGO 2.0, Beijing Genomics Institute, Shenzen, China).

### 2.6. Statistical Analysis

The results were presented as mean ± standard error of means (SEM), and analyzed using the One-way analysis of variance (ANOVA). Value $p < 0.05$ was considered as statistically significant.

## 3. Results

### 3.1. Antihyperglycemic Activity of FC Extract

The fasting blood glucose (FBG) level of normal rats was in a range between 4.4 ± 0.10 and 4.9 ± 0.20 mmol/L throughout 21 days of experiment, as shown in Figure 1. While the induction of STZ increased, the FBG level of normal rats was from 4.9 ± 0.20 mmol/L to a diabetic stage as resembled by the negative control of 23.7 ± 0.89 mmol/L. Both treatments of 400 mg/kg FC and 300 mg/kg metformin significantly decreased ($p < 0.05$) the FBG level of diabetic rats compared to the negative control (Figure 1). Diabetic rats administered with FC extract showed the reduction of the FBG level from 23.6 ± 2.87 to 12.6 ± 0.66 mmol/L, and it was almost similar to metformin that reduced FBG levels from 24.4 ± 3.27 to 13.22 ± 0.81 mmol/L.

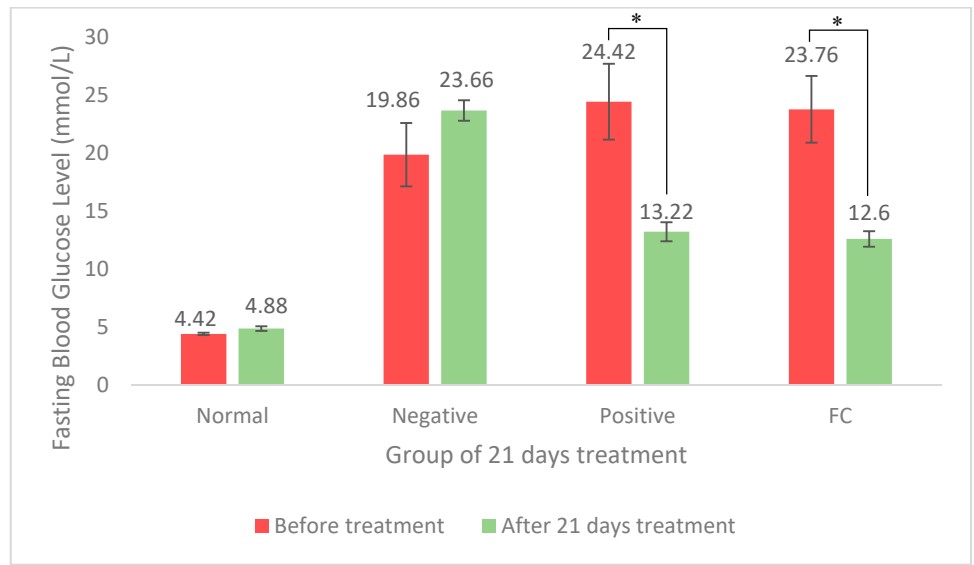

**Figure 1.** Fasting blood glucose level of rats from normal (normal group with distilled water), negative (diabetic group with distilled water), positive (diabetic group with 300 mg/kg metformin), and *Ficus carica* (FC) (diabetic group with 400 mg/kg *Ficus carica* aqueous extract). Symbol "*" represents significant difference ($p < 0.05$).

## 3.2. Sperm Quality Analysis

The parameters used for sperm quality assessment were sperm count, motility, the percentage of viability, and percentage of normal morphology. Figure 2 shows the comparison of sperm counts for normal control, negative control (diabetic without treatment), positive control (diabetic treated with metformin), and FC (diabetic with FC treated). Induction of diabetes with STZ significantly reduced ($p < 0.05$) the sperm count of normal control from $(88.6 \pm 6.85) \times 10^6$ sperm to $(2.8 \pm 1.36) \times 10^6$ sperms after 21 days of treatment. However, treatment with FC significantly increased ($p < 0.05$) the number of sperms to $(69.8 \pm 9.81) \times 10^6$ compared to negative control, and it was considerably higher than metformin which was $(14.0 \pm 3.65) \times 10^6$ sperms.

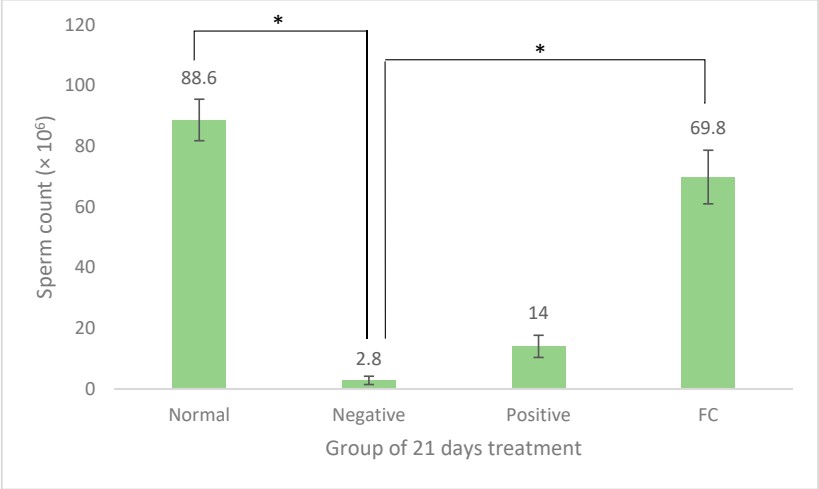

**Figure 2.** Sperm count ($\times 10^6$) of rats from normal (normal group with distilled water), negative (diabetic group with distilled water), positive (diabetic group with 300 mg/kg metformin), and FC (diabetic group with 400 mg/kg *Ficus carica* aqueous extract). Symbol "*" represents significant difference ($p < 0.05$).

Figure 3 shows the effect of FC treatment on the motility of diabetic rat sperm compared to the negative, positive, and normal control groups. Sperm motility refers to the activeness of sperm movement during microscopic observation which is based on WHO [21], either progressive motility, nonprogressive motility, or immotility. The induction of diabetes resulted in the increase of immotile sperm to the normal control from 25.52% ± 4.95% to 53.32% ± 6.98% as resembled by the negative control. Treatment with FC significantly increased ($p < 0.05$) the percentage of sperm with progressive motility in diabetic rats at 46.48% ± 2.06% compared to the negative control (6.80% ± 4.16%).

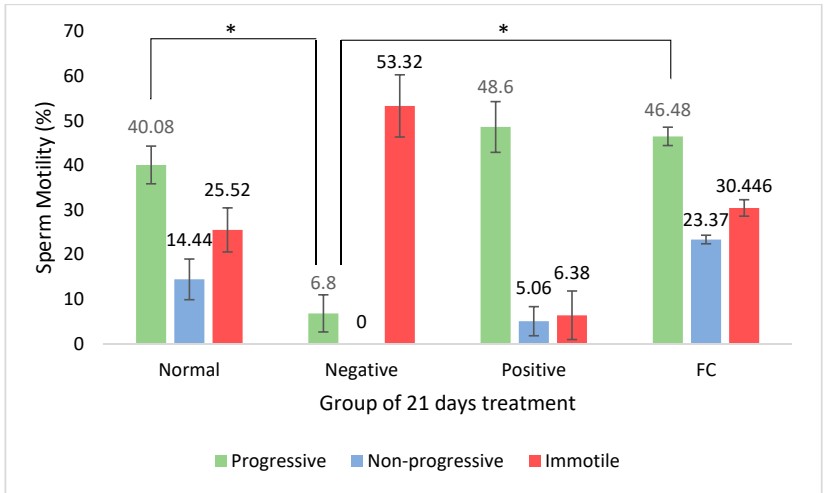

**Figure 3.** Sperm motility (%) of rats from normal (normal group with distilled water), negative (diabetic group with distilled water), positive (diabetic group with 300 mg/kg metformin), and FC (diabetic group with 400 mg/kg *Ficus carica* aqueous extract). Symbol "*" represents significant difference ($p < 0.05$).

The percentage of viability or percentage of live sperms for each group of rats in this study is shown in Figure 4. The percentage of sperm viability decreased significantly ($p < 0.05$) after the STZ induction, which was 6.68% ± 4.09%, compared to normal control (54.46% ± 4.41%). Diabetic rats treated with FC have shown a significant increase in percentage of sperm viability ($p < 0.05$) which was 69.91% ± 2.04% compared to the negative control (6.68% ± 4.09%), and higher than metformin (53.62% ± 7.4%).

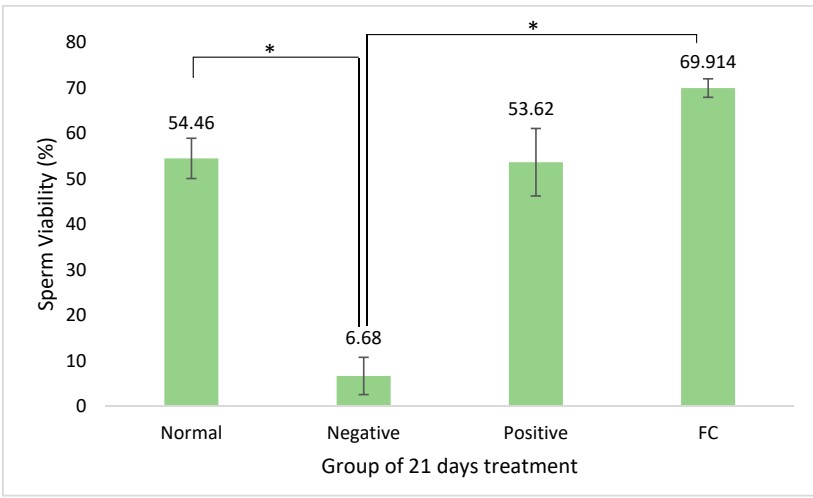

**Figure 4.** Sperm viability (%) of rats from normal (normal group with distilled water), negative (diabetic group with distilled water), positive (diabetic group with 300 mg/kg metformin), and FC (diabetic group with 400 mg/kg *Ficus carica* aqueous extract). Symbol "*" represents significant difference ($p < 0.05$).

Normal morphology of a sperm was observed on the three main parts of the sperm, which are head, midpiece, and tail. Figure 5 shows the effect of FC on the percentage of normal morphology of a diabetic rat sperm compared to the controls. Diabetes induction decreased the percentage of normal sperm morphology significantly ($p < 0.05$) from 33.74% ± 3.58%, as shown by normal control to 5.05% ± 1.43% as resembled by the negative control. After receiving the FC treatment, the percentage of normal sperm morphology increased significantly ($p < 0.05$) which was 38.04% ± 3.65% compared to negative control, and was observed to surpass the metformin group (21.36% ± 5.18%) and normal control.

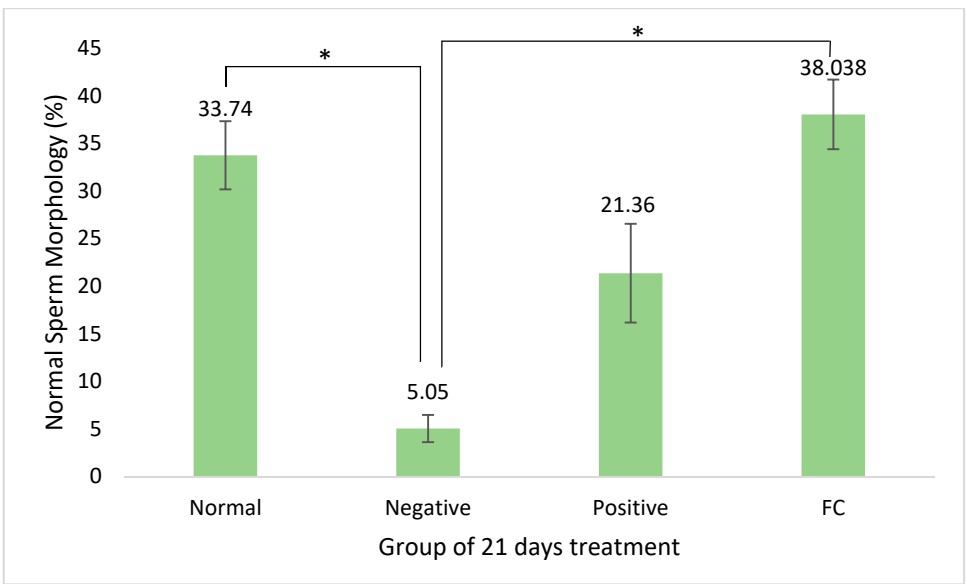

**Figure 5.** Normal sperm morphology (%) of rats from normal (normal group with distilled water), negative (diabetic group with distilled water), positive (diabetic group with 300 mg/kg metformin), and FC (diabetic group with 400 mg/kg *Ficus carica* aqueous extract). Symbol "*" represents significant difference ($p < 0.05$).

### *3.3. Sperm Proteomics Analysis*

### 3.3.1. Concentration of Sperm Protein

Table 1 shows the concentration of sperm protein (mg/mL) for each group determined by the Bradford assay [23]. The FC treatment group recorded the highest concentration of protein which was 1.15 mg/mL, followed by normal control 1.13 mg/mL, positive control 1.05 mg/mL, and negative control with the lowest concentration of 0.8 mg/mL.

**Table 1.** Concentration of sperm protein (mg/mL).

| Group | Concentration of Protein (mg/mL) |
|---|---|
| Normal | 1.13 |
| Negative (Diabetic without treatment) | 0.8 |
| Positive (Diabetic treated with metformin) | 1.05 |
| FC (Diabetic treated with *F. carica* extract) | 1.15 |

### 3.3.2. Sperm Protein Profile via SDS-PAGE

Based on Figure 6, the negative control group showed a thin protein band similar to positive control, whereas the FC group showed thicker protein band compared to negative control, especially at molecular weight of 9–19 kDa (yellow box) and 38–46 kDa (red box). In addition, the FC treatment group also showed thick protein band similar to normal control, indicating that the treatment might improve the fertility of diabetic male rats.

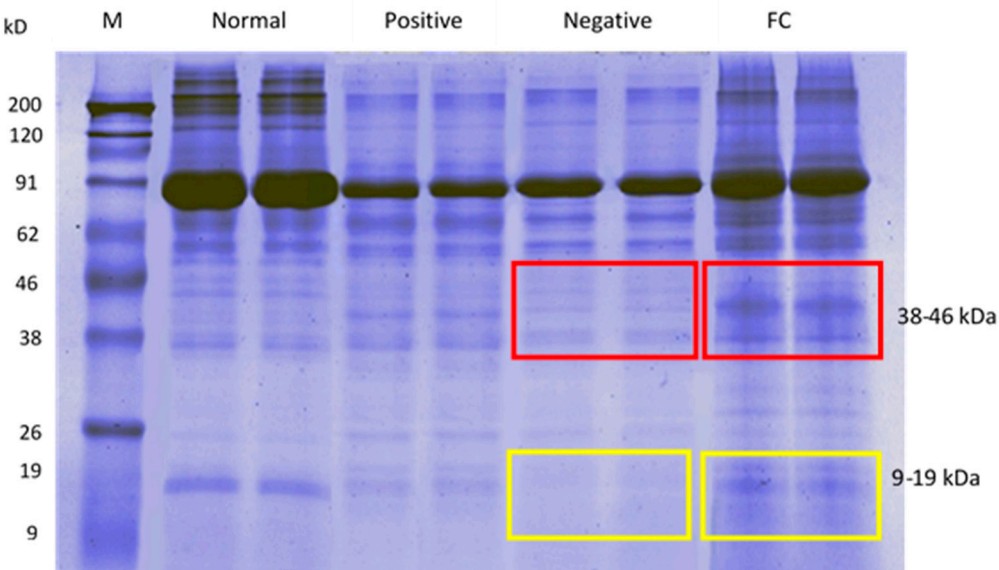

**Figure 6.** Sperm protein bands from four groups using SDS-PAGE electrophoresis. Column M is a molecular weight marker; normal: Sperm protein from normal group; positive: Diabetic treated with metformin; negative: Diabetic without treatment, and FC: Diabetic treated with *F. carica*.

### 3.3.3. LC-MS/MS Analysis

LC-MS/MS analysis recorded the total number of proteins present in each group (Tables S1–S4). According to Table 2, the sperms of negative control rat showed lower protein expression which was 55 proteins compared to normal control that showed high protein expression with 149 proteins. The metformin group recorded 90 sperm protein expressions, which was higher than negative control. Meanwhile, the FC group recorded almost a similar number of proteins as normal control which was 155 proteins.

**Table 2.** The total number of proteins in the sperm sample of FC group and the controls as determined by LC-MS/MS analysis.

| Group | Total Number of Protein |
|---|---|
| Normal | 149 |
| Negative (Diabetic without treatment) | 55 |
| Positive (Diabetic treated with metformin) | 90 |
| FC (Diabetic treated with *F. carica* extract) | 155 |

### 3.3.4. Gene Ontology Analysis via Blast2GO and WEGO

Based on gene ontology analysis via Blast2GO and WEGO, sperm proteomic analysis was focused on the reproductive process in order to evaluate the protein activity and function involved in diabetic rat fertility, as shown in Figure 7 and Figure S1. Expression of reproductive protein in normal control was 24 proteins, negative control was 11 proteins, positive control was 15 proteins, and FC group with the highest number of 34 proteins.

### 3.3.5. Sperm Unique Protein upon FC Treatment

Based on the Uniprot database, approximately 32.3% of total sperm proteins in the FC treated group have been identified on fertility-related function and activity. Table 3 shows a total of 14 unique fertility proteins in FC group (negative control did not have these proteins) were identified along with their fertility-related function and molecular weight. The proteins were then classified into two groups according to their functions in the fertility of diabetic rat. The two groups of proteins were

protective unique protein (manganese superoxide dismutase, peroxiredoxin-5, endoplasmic reticulum stress regulator ATPase, pyruvate kinase, ATP synthase subunit O and pyruvate dehydrogenase E1 component subunit alpha) and reproductive unique protein (myosin-9, prostaglandin-H2 D-isomerase, histone H2B, dynein light chain 1, ropporin-1, T-complex protein 1 subunit beta, dihydrolipoyl dehydrogenase, and Izumo sperm-egg fusion protein 1). The classification of reproductive unique protein was based on the functions of spermatogenesis and fertilization, as shown in Table 3. The unique protein molecular weight data was then matched back to the FC group protein bands in SDS-PAGE gel to show the distribution of this fertility protein according to molecular weight (Figure 8).

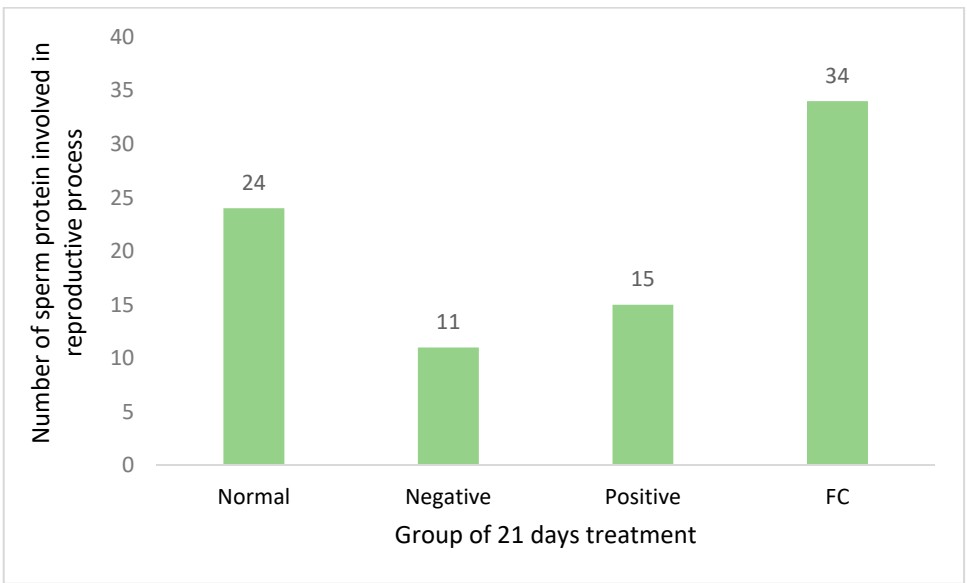

**Figure 7.** The number of sperm protein involved in reproductive process from normal (normal group with distilled water), negative (diabetic group with distilled water), positive (diabetic group with 300 mg/kg metformin), and FC (diabetic group with 400 mg/kg *Ficus carica* aqueous extract).

**Table 3.** Classification of 14 unique sperm proteins of the FC treatment group compared to negative controls according to function based on Uniprot database.

| Protein ID (Uniprot) | Name | Location | Molecular Weight (kDa) |
|---|---|---|---|
| *1. Protective unique protein* | | | |
| P07895 | Manganese superoxide dismutase | mitochondria | 24.66 |
| Q9R063 | Peroxiredoxin-5, mitochondrial | mitochondria | 22.17 |
| P46462 | Transitional endoplasmic reticulum ATPase | nucleus | 89.29 |
| P11980 | Pyruvate kinase PKM | cytoplasm | 57.78 |
| Q06647 | ATP synthase subunit O | mitochondria | 23.38 |
| Q06437 | Pyruvate dehydrogenase E1 component subunit alpha | mitochondria | 43.37 |
| *2. Reproductive unique protein* | | | |
| i. Spermatogenesis | | | |
| Q62812 | Myosin-9 | cytoplasm | 226.2 |
| P22057 | Prostaglandin-H2 D-isomerase | nucleus | 21.29 |
| Q00729 | Histone H2B type 1-A | nucleosome | 14.22 |
| P63170 | Dynein light chain 1 | nucleus | 10.36 |
| ii. Fertilization | | | |
| Q4KLL5 | Ropporin-1 | cytoplasm | 23.95 |
| Q5XIM9 | T-complex protein 1 subunit beta | cytoplasm | 57.42 |
| Q6P6R2 | Dihydrolipoyl dehydrogenase | mitochondria | 54.0 |
| Q6AY06 | Izumo sperm-egg fusion protein 1 | acrosome | 43.55 |

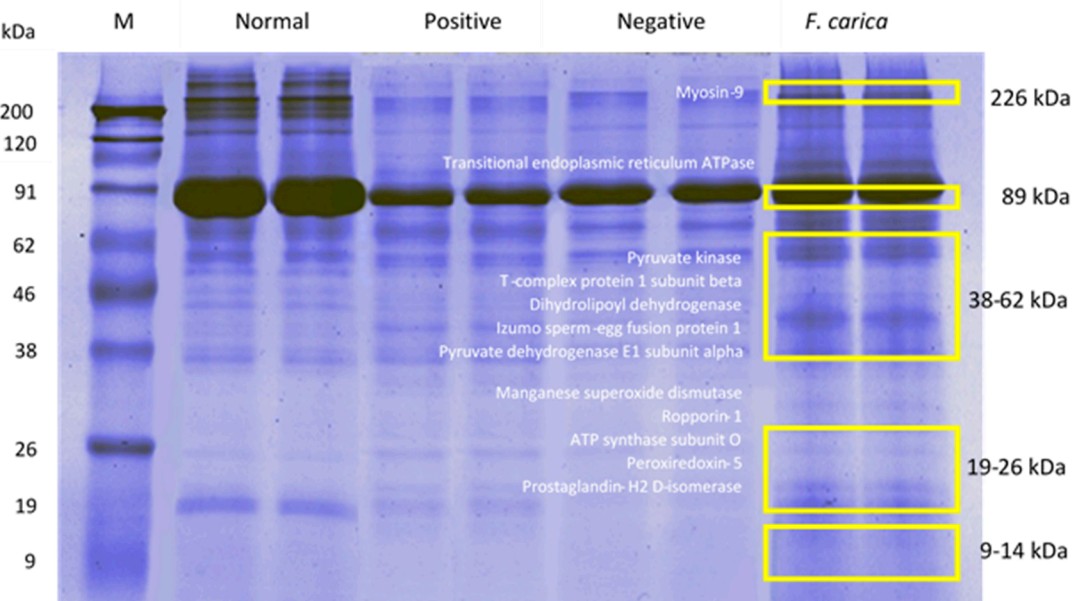

**Figure 8.** The distribution of unique fertility-related protein of FC treated sperm group based on molecular weight in SDS-PAGE.

## 4. Discussion

### 4.1. Antihyperglycemic Activity of FC Extract

STZ [2-deoxy-2-(3-(methyl)-3-nitrosoureido)-D-glucopyranose] is an antimicrobial and alkylated chemotrapeutic agent synthesized by *Streptomycetes achromogenes* [25]. Rakieten et al. [26] reported that STZ was diabetogenic when it caused diabetes via the specific necrosis of β-cells located at *islet of Langerhans* in the pancreas. King [25] describes the mechanism of diabetic induction by STZ, in which it enters β-cells through Glut-2 carrier protein (SLC2A2), and causes the alkylation of DNA. DNA alkylation activates poly (ADP-ribose) polymerase (PARP) that results in depletion of nicotinamide adenine dinucleotide (NAD+) and adenosine triphosphate (ATP), in which both NAD+ and ATP are needed for insulin secretion [27]. Therefore, the depletion of NAD+ and ATP leads to inhibition of insulin production. Pancreatic β-cells produce insulin in order to maintain normal blood glucose levels. Insulin activates glycogen synthase that converts glucose to glycogen, and also activates carrier protein GLUT4 which carries glucose from the blood stream into cells [28]. Thus, when there are problems related to insulin production, the conversion of glucose to glycogen and glucose uptake by the cell will be affected, resulting in the increase of blood glucose level.

Duca et al. [29] reported that the mechanism of blood glucose level reduction by metformin is by decreasing hepatic glucose production via inhibition of mitochondrial complex I, which results in the increase of adenosine monophosphate-activated protein (AMP) and the activation of adenosine monophosphate-activated protein kinase (AMPK). AMPK deactivates the transcription of peroxisome proliferator-activated receptor gamma (PPAR-γ) and sterol regulatory element binding protein 1c (SREBP-1c), in which both of them are regulators in the expression of enzymes involved in gluconeogenesis, glucose transportation, and fatty acid synthesis in the liver [30]. The inhibition of these enzyme expressions eventually causes reduction of blood glucose level.

Meanwhile, FC reduces blood glucose level by facilitating the glucose uptake by the muscle cell, which is the similar action such as insulin in reducing blood glucose level [18]. Nearly 40% of total body weight is represented by the skeletal muscle, in which it is also an important tissue that has been targeted for insulin action and glucose uptake [31]. In vitro experiment done by Perez et al. [18] found that there was a low glucose uptake by skeletal muscle of untreated diabetic rats due to the insufficient amount of insulin, but on the contrary, diabetic rats treated by FC showed a significant

increase of glucose uptake by skeletal muscle. Irudayaraj et al. [16] suggest that FC is able to improve the sensitivity of insulin, possibly due to the improvement of defects in insulin function that has been realized after a significant reduction of blood glucose and normalization of plasma insulin levels were found. Mopuri et al. [32] reported that FC also inhibits the activity of digesting enzymes of carbohydrates such as α-amylase and α-glucosidase. The inhibition of these enzymes may slow down the digestion of carbohydrates, thus leading to low glucose levels in the blood stream. Previous studies of some antihyperglycemic plants such as *Gynura procumbens* [14] and *Moringa olifeira* [33] also demonstrated similar mechanism as FC, via the increase of peripheral glucose uptake and inhibition of α-amylase and α-glucosidase. Meanwhile, there were other plants such as *Aloe vera* [34] and *Camillia sinensis* [35] that showed different antihyperglycemic mechanism compared to FC which were an improvement of insulin secretion and pancreatic β-cell function.

Therefore, all studies stated above indicate that there are phytochemicals in FC responsible for its antihyperglycemic property. Zhang et al. [36] conclude that the antihyperglycemic activity of FC may be due to the antioxidant effect of its phytochemicals. Certain antioxidants in FC that may contribute to its antihyperglycemic effect are quercetin [2,37,38], kaempferol [37,39], ficusin [40], ferulic acid [2], and caffeoylmalic acid [39]. It is found that quercetin delays oxidant injury and cell death due to the oxidation by free radicals [37], improves insulin signaling and sensitivity [41], as well as increases glucose uptake by cell [42]. Kaempferol inhibits the breakdown of polysaccharides to glucose by α-glucosidase, as it is a carbohydrate digesting enzyme [43], and activates AMPK to mediate an antidiabetic effect by inhibiting hepatic glucose production [30]. Ficusin restores insulin level in diabetic rats due to regularized β-cell, and improves glucose utilization by increasing the expression level of glucose transporter protein (GLUT) [40]. Next, ferulic acid inhibits gluconeogenesis via the decrease of enzyme of gluconeogenesis, which are phosphoenolpyruvate carboxinase (PEPCK) and glucose-6-phosphatase [44]. Meanwhile, Takahashi et al. [39] found that caffeoylmalic acid is among the abundant polyphenol present in FC that has high antioxidant activity similar to vitamin C.

## 4.2. Sperm Quality Analysis

Conventional sperm quality analysis such as sperm count, motility, viability, and morphology using haemocytometer is still a relevant, effective, and fundamental investigation method that is able to detect the sign of male infertility as recommended by World Health Organization [45,46]. The sperm count from ejaculation is an early indicator towards the ability of the testis to produce sperm, as normal ejaculation is correlated with testicular volume [21]. A high percentage of progressive motility is important in determining the success of sperms to find and penetrate oocyte during the fertilization process, because the life span of a sperm is short, which is only 6–12 h after ejaculation [47]. Sperm viability is considered as an early indicator for male fertility, as it indicates either the sperms live or die [48]. While normal sperm morphology is crucial to determine the success of fertilization [49], because defects related to head, midpiece, and tail commonly result in nonprogressive motility and immotility [48].

The results showed that STZ-induced diabetic rats were infertile because the number of sperms, the percentage of sperm viability, and normal sperm morphology percentage were at the lowest, and the percentage of immotile sperm recorded was the highest. STZ causes the necrosis of β-cell in the islet of Langerhans in the pancreas which causes insulin production failure [26]. Based on a study by Kim and Moley [50], insulin deficiency causes failure of transcription and activation of GLUT glucose transport proteins. GLUT plays an important role in transporting glucose into sertoli cells to carry out the glycolysis process, which later produces lactate [51]. Lactate is a source of energy that is important in the growth of germ cell and sperm maturation [52]. When lactate fails to be supplied for germ cell nutrition, the spermatogenesis process is affected, and results in the decrease of sperm quality. According to Pavlinkova [4], diabetes decreases sperm concentration, viability, and increases sperm apoptosis due to the changes in protamine 1/protamine 2 ratio, small proteins that bind sperm DNA which are important for DNA stability and sperm maturation. Furthermore, the decrease in

sperm quality is closely related to oxidative stress due to hyperglycemia, which causes the excessive production of free radical in the blood such as reactive oxygen species (ROS) [53]. Ding et al. [6] found that high concentration of free radicals induced sperm apoptosis, sperm DNA fragmentation, and changed the expression of fertility proteins, in which they act as antioxidants to protect cells from structural modification by advanced glycation end (AGE) products.

In this study, metformin was found to increase sperm quality parameters, but it was not significant when compared to the negative control. According to Owen et al. [12], metformin increases sperm quality parameters by blocking complex I in the mitochondrial respiratory chain, resulting in the decrease of mitochondrial function and cell respiration, and it also causes the increase of AMPK which eventually leads to anaerobic respiration and the increase of lactate production. Lactate is the main source of energy for germ cells in order to be developed into mature sperms. Bertoldo et al. [11] state that metformin increases the level of mRNA for glucose carrier Glut 1 and lactate dehydrogenase which catalyzes the conversion of pyruvate into lactate. However, some studies have found that metformin lowers male fertility parameters. Research done by Naglaa et al. [15] using alloxan-induced diabetic rats shows a decrease in testicular weight, sperm count, sperm motility, and increase in the number of abnormal sperm when treated with metformin. Furthermore, the study by Hurtado de Llara et al. [13] shows that metformin inhibits sperm mitochondrial membrane potential and motility of boar sperms.

Based on the results of the study, the FC extract was able to improve the quality of sperms in diabetic rats. Since there is no study that had been done for male infertility due to diabetes using FC leaves extract as treatment, we considered choosing the concentration of FC from an antidiabetic study such as Irudayaraj et al. [16], Jayakumar et al. [54], and Mopuri et al. [32] (that used concentration between 400–500 mg/kg) because we believe that the problem is due to diabetes. We also considered the concentration of FC from studying the profertility of FC by Naghdi et al. [17] and Shimaa et al. [55] (that used concentration of FC extract between 200–300 mg/kg), where the concentration was below than what antidiabetic studies used. So based on these studies, we chose 400 mg/kg FC as it was the concentration used in antidiabetic studies and we estimated that the concentration can as well give a good result for profertility. The ability of FC extract to treat diabetes, as well as fertility is closely related to its phytochemical content that acts as an antioxidant in reducing reactive oxygen species production (ROS) [55,56]. Takahashi et al. [39] reported that caffeoylmalic acid is among the most abundant polyphenol found in FC which has the same level of antioxidant activity as vitamin C. The increase in sperm quality parameters is also likely to be due to the positive effect of FC on the hormone involved in male fertility. FC contains saponin which is found to resemble testosterone action, increase testosterone production, and has good effects on follicle stimulating hormone (FSH) [55]. These positive effects accelerate sperm maturation and sperm count. Furthermore, Irudayaraj et al. [16] reported that FC normalizes the plasma insulin levels of diabetic rats. Kim and Moley [50] reported that insulin plays an important role in male reproductive system when the diabetic mice treated with insulin show a significant increase of sperm count and motility compared to diabetic mice without treatment. Mice treated with insulin also show increase of glucose transporter (GLUT) protein expression in leydig and sperm cells. The GLUT protein functions to carry glucose into testicular cells for the continuation of spermatogenesis. Active spermatogenesis activity increases sperm quality parameters such as sperm count, motility, normal morphology, and viability [57].

*4.3. Sperm Proteomics Analysis*

4.3.1. Sperm Protein Concentration, Protein Expression via SDS-PAGE, LC-MS/MS, and Gene Ontology Analysis

It was consistent that the negative control showed lowest reading for all proteomic parameters; protein concentration, protein band thickness, and total number of proteins. The decrease of the parameters by negative control may be due to the effect of diabetes on endoplasmic reticulum in the sperms. The endoplasmic reticulum is a site of synthesis, folding, and maturation of proteins, which is present in any cell including the sperm [58]. Diabetes and hyperglycemia result in the increase

of endoplasmic reticulum stress [59], which causes changes in the homeostasis of the endoplasmic reticulum, and subsequently reduces protein synthesis [60]. Previous studies also supported our findings [61,62]. An et al. [10] reported that no expression of the spermatogenesis regulator protein was recorded in the sperm of diabetic patients compared to normal. Decreased expression of other proteins in the sperm of diabetic patients was also reported by Kim and Moley [50] and Kriegel et al. [61]. Based on gene ontology analysis, diabetes also causes a decrease in the expression of reproductive protein. Gene ontology analysis is a recent development in the molecular biology study that enables genes or proteins to be classified based on their function and involvement in any biological process, hence facilitates scientists to understand the mechanism behind the phenotype [63]. Our study focused on reproductive proteins out of all biological processes since reproduction is an important process that determines fertility. In correlation with sperm quality analysis in our study, it can be speculated at this point that diabetes may reduce sperm quality by depleting the production of reproductive protein.

Interestingly, FC treatment improved all proteomic parameters of diabetic rats. Furthermore, gene ontology analysis revealed that FC treatment increased the number of sperm reproduction protein despite being in a diabetic condition. The increase of the sperm reproduction protein number was in correlation with the increase of sperm quality parameters, as shown in Figures 2–5. It can be inferred that sperm fertility proteins influenced by FC protect male fertility from diabetes complication. The study using another herb by Ghosh et al. [64] found that treatment with *E. jambolana* increases the expression of the protein Apoptosis regulator (BCL2) in the testicular tissue of diabetic mice. BCL2 is a protein that inhibits the process of apoptosis and inflammation of cells [65]. With increased expression of BCL2 protein, damage to testicular cells can be avoided, thus protecting male fertility from diabetes complication.

### 4.3.2. Sperm Unique Protein upon FC Treatment

Proteomic analysis later was focused on the unique sperm fertility protein of the FC treated group in comparison with the negative controls. This investigation aimed to evaluate how the unique fertility proteins escape diabetes complication and thus maintain male fertility. The heaviest unique protein was myosin-9 with a molecular weight of 226.2 kDa, followed by a transitional endoplasmic reticulum ATPase with a molecular weight of 89.29 kDa. The protein bands between 38 and 62 kDa contained pyruvate kinase protein, T-complex protein 1 subunit beta, dihydrolipoyl dehydrogenase, Izumo sperm-egg fusion protein 1, and pyruvate dehydrogenase E1 alpha subunit. Furthermore, the protein bands between 19 and 26 kDa contained manganese superoxide dismutase, ropporin-1, ATP synthase subunit O, peroxiredoxin-5, and prostaglandin-H2 D-isomerase, while the lowest molecular weight proteins were histone H2B and dynein light chain, both of them in the protein bands between 9 and 14 kDa.

Based on protein molecular weight, many previous studies have concluded that low-molecular weight proteins have high antioxidant activity [66–68]. An in vitro study by Pierro et al. [67] found that the process of hydrolysis of cow's casein by latex FC produced low molecular weight proteins with high antioxidant activity. The LC-MS/MS analysis performed in this study also support the statement, in which we actually found that the rat sperm of FC contained manganese superoxide dismutase (MnSOD) and peroxiredoxin-5 proteins, and both have high antioxidant activity and also low molecular weight proteins (<25 kDa). MnSOD and peroxiredoxin-5 may act to prevent ROS increase in sperm cells, and protect cell organelles, especially mitochondria, which play a key role in ensuring progressive sperm motility as it functions to produce energy in the cell. Manganese superoxide dismutase (MnSOD) is a major antioxidant defense in mitochondria [69]. MnSOD is an enzyme that converts superoxide anion radicals to hydrogen peroxide, thereby reducing the probability of superoxide anions interacting with nitric oxide to form reactive nitrite peroxides [70]. The study by Li et al. [71] found that infants with MnSOD deficiency are genetically exposed to mitochondrial damage in several organs including the heart, causing death within 10 days after birth. This indicates that MnSOD has an important role in the mitochondrial defense system. MnSOD expression in diabetic rats has been shown to suppress

the increase of ROS, and inhibit the effect of hyperglycaemia [72–74]. Additionally, Cocchia et al. [75] reported that MnSOD protects spermatozoa from damage by lipid peroxidation, and subsequently increases sperm motility and viability compared to the negative control. Meanwhile, peroxiredocin-5 is also an effective antioxidant enzyme in the inhibition of peroxide in mitochondria. The uniqueness of the peroxiredoxin family compared to other peroxidases is due to the hydrogen peroxide catalysis initiated by the peroxidatic cysteine located at the N-terminus of the protein [76]. Peroxidatic cysteines break down the O-O bond of the hydrogen peroxide which then temporarily produce cysteine sulfenic acid, before returning to its original form of peroxidatic cysteine in order to catalyze other hydrogen peroxide [77]. Kubo et al. [78] reported that the expression of peroxiredoxin-5 protein is found to be responsible for the suppression of apoptosis in pig pericyte cells when the protein is able to reduce oxidative stress induced by diabetes. In addition, O'Flaherty [79] states that peroxiredoxin plays a key role in the protection of spermatozoa function from ROS when it is found that peroxiredoxin deficiency in infertile men is consistent with a decrease in sperm quality.

Another protein, the transitional endoplasmic reticulum ATPase protein or known as the valocin-containing protein (VCP) had the function in regulating endoplasmic reticulum stress induced by diabetes. Excessive endoplasmic reticulum stress induced by diabetes causes alterations in the homeostasis of the endoplasmic reticulum and subsequently reduces protein synthesis [59,60]. In addition, excessive endoplasmic reticulum stress is linked with male infertility [80]. Guzel et al. [81] state that spermatogenesis requires intensive protein synthesis to continue the developmental process from spermatogonia to spermatozoa, however, uncontrolled endoplasmic reticulum stress inhibits spermatogenesis. VCP is a family of ATPases that generally functions in protein folding or unfolding [82]. Additionally, the specific function of VCP is to regulate the degradation of unfolded proteins in response to endoplasmic reticulum stress [83]. Wojcik et al. [84] reported that disruption of VCP RNA results in the reduction of VCP protein expression in cells, thus inducing the changes on over 30 RNA transcripts for other proteins involved in endoplasmic reticulum stress, amino acid starvation, and apoptosis. Hence, the VCP protein presence in FC diabetic rats retains the normal functioning of the endoplasmic reticulum, and thus ensures the continual production of proteins required for sperm development despite being in a diabetic condition.

There are three unique proteins in the sperm of FC group involved in glycolysis and the production of ATP or Nicotinamide adenine dinucleotide (NADH), which are pyruvate kinase, pyruvate dehydrogenase E1 component subunit alpha, and ATP synthase subunit O. Previous studies reported that diabetes disrupts the expression of the proteins in the early pathogenesis and later causes severe complications [85–87]. Disruption of ATP synthase by diabetes stimulate mitochondrial ROS generation and thus promote severe complications on the tissue and organ [85]. The failure of diabetes to disrupt the proteins at the beginning of pathogenesis will terminate the complication of diabetes [86,87]. Qi et al. [86] reported that the activation of pyruvate kinase may protect against diabetes by improving glucose metabolism, inhibiting the production of toxic glucose metabolites and inducing the biogenesis of mitochondria. Another study done by Rahimi et al. [87] concluded that activation of pyruvate dehydrogenase during a diabetic condition reduces available substrate for production of glucose, resulting in low blood glucose level and improved glucose tolerance. It is inferred that treatment with FC expressed these unique proteins thus protecting the sperm from toxicity of diabetes.

The following unique proteins are fertility proteins involved directly in the reproduction process; spermatogenesis (myosin-9, prostaglandin-H2 D-isomerase, histone H2B, and dynein light chain 1) and fertilization (ropporin-1, T-complex protein 1 subunit beta, dihydrolipoyl dehydrogenase, and Izumo sperm-egg fusion protein 1), where the expression of these reproductive unique proteins may be due to the protective effect of the unique proteins discussed before (MnSOD, peroxiredoxin-5, VCP, ATP synthase, pyruvate kinase, and pyruvate dehydrogenase) against diabetes complication.

Myosin-9 functions as a component in the activation of FSH hormone synthesis. The study by Lin et al. [88] reported that myosin-9 is involved in the activation of FSH hormone synthesis by the

guanosine triphosphate (Gαh) protein. It was found that myosin-9 inhibition using Blebbistatin, a type of ATPase, inhibited the activation of FSH hormone synthesis by Gαh protein in Sertoli cells. FSH hormone secretion by myosin-9 induces FSH activity in the development of spermatogonia to spermatocytes. The next protein, prostaglandin-H2 D-isomerase was involved in Sertoli cell differentiation. Prostaglandin-H2 D-isomerase or known as prostaglandin D2 synthase (PGDS) is an enzyme that converts the cyclooxygenase product of prostaglandin H2 (PGH2) to prostaglandin D2 (PGD2) [89]. PGD2 is a protein that enhances the differentiation of Sertoli cells, and controls the proliferation of germ cells [90]. The study by Moniot et al. [91] states that expression of the Sox9 gene, which is a transcriptional factor of differentiation of Sertoli cells, is regulated by the PGDS protein. Whereas, Samy et al. [92] reported that increased expression of PGDS was consistent with the increase in germ cell tight junction, indicating that there is a correlation between PGDS expression and junction. The interplay between germ cells, which is part of the blood-testis structure, is important in regulating the molecular entry and development of germ cells [51]. There are two other unique FC proteins identified that help in spermatid development and maturation of spermatozoa which are histone H2B and dynein light chain 1. Histone H2B (HH2B), which is a variant of histone protein, together with DNA binding to nucleosomes in chromatin [93], was known to be involved in spermatid development. Lu et al. [94] reported that the highest HH2B expression was recorded in spermatogonia, followed by spermatocytes and finally spermatids. This implies that HH2B plays a role in every stage of sperm cell in spermatocytogenesis. Dynein light chain 1 (DLC1) is a protein involved in the spermatozoa maturation process. Dynein generally acts to regulate chromosome movement, assembly, and orientation of mitotic spindles, as well as nucleus migration [95]. The study by Wang et al. [96] in rat sperm suggests that DLC1 plays a role in chromatin condensation in the early stages of spermatids, disposal of excess cytoplasm for spermatozoa formation, and release of spermatozoa from the apical compartment into the lumen of the seminal tuberculosis (sperm) when there is intensive DLC1 expression in the spermatid nucleus length and cytoplasm of spermatozoa.

In this proteomic study, a unique sperm protein from FC group was identified to be responsible for fibrous sheath signaling that subsequently influenced sperm motility. The protein is ropporin-1 (ROPN1) which is found only in fibrous sheath of the sperm flagella, and is specifically located at the center and tip of the flagella [97]. ROPN1 binds to the primary components of fibrous sheath, the A-kinase anchoring proteins (AKAP) 3 and 4 to initiate the signaling pathway for sperm motility [98]. Fiedler et al. [98] found that a rat sperm lacked the ROPN1, and ropporin-1 like protein (ROPN1L) had problems with flagella and became immotile. This indicates that ROPN1 is a vital protein in sperm motility and capacity. The discovery of this protein in the sperm of FC group is in line with the results of sperm motility analysis which has shown that treatment with FC helps increase sperm motility in diabetic rats. There is a unique sperm protein in the FC treated group that is known to have activity that helps the binding of the sperm to ZP which is T-complex protein 1 subunit beta. The T-complex protein 1 subunit beta (CCT) is a chaperone-type protein that is commonly known to fold other proteins from its original or nonfoldable state to a folded and functional form [99]. While in fertility, CCTs are the first isolation in mice testes protein that are specifically involved in sperm motility, sperm capacitation, and the ability to bind and penetrate ZP [100]. A proteomic study of CCT by Dun et al. [101] using blue native polyacrylamide gel electrophoresis (BN-PAGE) on spermatozoa found high CCT expression on the surface of the spermatozoa through the capacitation process, while the far Western Blot method found that CCT showed adherence to intact ZP. This clearly indicates that CCT plays an intermediate role in the binding between sperm and ZP. Next, dihydrolipoyl dehydrogenase is a protein identified in the sperm of the FC group which is involved in the acrosome reaction process. Dihydrolipoyl dehydrogenase (DLD) is a post-metabolism pyruvate enzyme, a pyruvate dehydrogenase complex subunit E3 that is involved in the regulation of lactate in spermatozoa [102]. DLD has been reported to play a role during hyperactivation, acrosome reactions [103] and it is also involved in fetal development [104]. A study by Mitra and Shivaji [105] found that the downregulation of DLD activity by inhibitors, specifically 5-methoxyindole-2-carboxylic acid, prevented acrosome

reaction and decreased spermatozoa hyperactivation. During adhesion between the sperm membrane and the egg membrane, there are various protein interactions that occur. Sperm proteomic studies have identified that the unique protein of the FC group, Izumo sperm-egg fusion protein 1 is involved in sperm-egg fusion. Izumo sperm-egg fusion protein 1 (IZUMO1) is a protein that is specifically involved in sperm-egg binding and fusion. The protein is identified by Inoue et al. [106] using monoclonal antibodies that inhibit sperm-egg fusion and gene cloning. It is not detectable on the surface of newly released spermatozoa (after ejaculation or in the epididymis), however, the protein can only be detected after acrosome reaction. This is probably because IZUMO1 is hidden inside the plasma membrane and only present on the surface as a result of the acrosome reaction. Inoue et al. [103] also found that mice that lack the IZUMO1 gene have normal sperm morphology and are capable of binding and penetrating ZP, however, they are unable to proceed fusion with the egg. A further study by Bianchi et al. [107] found that IZUMO1 binds to its receptor, namely JUNO protein, located on the surface of the egg, and then these two proteins interact with each other, which subsequently activates sperm-egg fusion.

Therefore, we proposed that FC treatment induces the expression of protective unique proteins that prevent the complication of diabetes on the reproductive unique protein, and thus resulted in the amelioration of male fertility despite being in a diabetic condition.

## 5. Conclusions

Overall, leaf extract of *Ficus carica* (FC) shows antihyperglycemic and profertility effects on streptozotocin-induced diabetic male rats. Treatment with FC extract shows a significant decrease in blood glucose level closer to normal. These antihyperglycemic effects have been found to prevent the complications of diabetes on male rat fertility in accordance with the recovery of sperm quality parameters after treatment with FC extract. This recovery and protection may be due to the bioactive compounds responsible for the antioxidant activity present in FC extracts such as quercetin, kaempferol, ferulic acid, and caffeoylmalic acid. In addition to acting in order to balance oxidative stress caused by diabetes, these bioactive compounds may also be involved in the secretion of insulin, proteins, and other molecules leading to fertility improvement.

Proteomic studies on the sperm of the FC group show an increase in overall protein expression compared to controls. Furthermore, the comparison between FC group proteins and negative control reveals that the sperm of FC group contains unique proteins that have fertility-related functions. These unique FC group proteins are made up of protective and reproductive unique proteins. The synergy interaction between these unique proteins may prevent the complications of diabetes on the function of testes and sperm cells, and thus restore the fertility of diabetic male rats.

**Supplementary Materials:** The following materials are available online at http://www.mdpi.com/2227-9717/8/4/395/s1, Table S1: List of sperm proteins in normal control group; Table S2: List of sperm proteins in negative control group; Table S3: List of sperm proteins in positive control group; Table S4: List of sperm proteins in *Ficus carica* treated group; Figure S1: Analysis of sperm proteomic data from (a) normal, (b) negative control, (c) positive control, and (d) *Ficus carica* treated group using the WEGO software.

**Author Contributions:** Conceptualization, U.A.B., P.S., and M.M.N.; methodology, M.M.N., K.A.K., N.A.K.B., A.K., M.J., W.M.A., and M.S.; software, W.M.A.; validation, M.M.N., W.M.A., and M.J.; formal analysis, U.A.B.; investigation, U.A.B., P.S., N.A.K.B., and A.K.; resources, M.S., M.J., W.M.A., and M.M.N.; data curation, W.M.A. and U.A.B.; writing—original draft preparation, U.A.B.; writing—review and editing, U.A.B., W.M.A., and M.M.N.; visualization, U.A.B. and M.M.N.; supervision, M.M.N., M.J., and W.M.A.; project administration, U.A.B., P.S., N.A.K.B., and A.K.; funding acquisition, M.M.N., M.J., and W.M.A. All authors have read and agreed to the published version of the manuscript.

**Funding:** This research has been funded by the Research University Grant (GUP-2016-056).

**Acknowledgments:** The authors wish to thank Universiti Kebangsaan Malaysia for providing the facilities in the Animal House, Faculty of Science and Technology, Universiti Kebangsaan Malaysia. Special thanks to the Mass Spectrometry Technology Section, Malaysia Genome Institute (MGI) for contributing in LC-MS/MS analysis for identification of the sperm protein. Last but not least, thank you to Saf Fa Fig Garden for providing *Ficus carica* leaves as the sample of the research.

**Conflicts of Interest:** The authors declare no conflict of interest.

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
