# Peer review of "Sperm Proteomics Analysis of Diabetic Induced Male Rats as Influenced by Ficus carica Leaf Extract"

_processes, doi:10.3390/pr8040395_

Round 1

Reviewer 1 Report

It is a large interesting study. However, the authors should more elaborate and mainly interconnect particular information, especially in the Introduction. It is not clear from the literature search what is the aim of study.
I strongly suggest to separate results and discussion. Although both these chapters are together, they are not absolutely interconnected. In the Results, there are missing references to supplementary material (tables and graphs).
First of all, the description of the importance of individual analyzed reproductive proteins needs to be linked in the relation to diabetes.

More particular comments:
1) l. 38-39  more information about diabetes and sperm parameters authors find in Pavlinkova et al. Sci Rep 2017
2) l. 58 citation is missing
3) l. 73-76 based on which studies - references
4) l. 94 On what basis was this concentration of FC chosen?
5) l. 109 please define Biggers-Whitten-Whittingham (BWW) medium
6) l.137 please specify the lysis buffer
7) l. 176 title of chapter is not exact
Graphs should always be placed below the text concerning with them
Figure 6. Which protein concentration did apply per each lane? May authors use any reference protein such as actin?
Authors should restructure discussion part and interconnect information about analyzed proteins with diabetes.

Reviewer 2 Report

Authors in their study demonstrate the anti-hyperglycemic and pro-fertility activities of the Ficus carica (FC) leaf extract in rat model of streptozotocin-induced diabetes. Authors demonstrate the FC leaf extract is able to improve the quality of sperm in diabetic rats by increasing the number of sperm, the percentage of sperm with progressive motility, the percentage of sperm viability, and the percentage of normal sperm morphology. Furthermore, based on the obtained results of sperm proteomics analysis, authors concluded that FC leaf extract improve the infertility of diabetic male rats by the recovery of the expression of several sperm proteins, involved in either antioxidant activity, endoplasmic reticulum stress regulation, glycolysis and energy synthesis, or reproduction. In overall, the present study is well written and easy to follow; the Material and Methods section contains all the necessary details to reproduce the experimental part of the study. The Introduction section, the Results and Discussion, and the Conclusions are adequate. The present study could be interesting to the broad audience in the field of medical biology as it raises the interesting possibility to use the natural leaf extract from herb together with the synthetic drugs (such as Metformin) to improve the quality of life of diabetic patients.

Major comments:

Sperm quality analysis (sperm count, motility, viability, and normal sperm morphology) in the present study was assessed under light microscope using haemocytometer or glass slides only. Neither flow cytometry nor computer assisted sperm analysis (CASA) were used to evaluate sperm quality in details. Lines 269 – 271: please, clarify (discuss briefly) the reason of the low percentage of spermatozoa with normal morphology in normal, non-diabetic rats.

Minor comments:

It is better to include the corresponding p-values, and to flag the levels of significance (by means of different symbols or stars) directly on the figures, not in the manuscript text only. References in the list of references should be given with abbreviated journal name (World J Dairy Food Sci, Indian J Physiol Pharmacol and so on).

Reviewer 3 Report

This is an interesting paper showing the effect of Ficus carica extracts, an herbal plant, on sperm parameters in diabetic rats. The aim of this study was to examine the possible protective effect of Ficus carica extracts on sperm concentration, viability, morphology, and protein expression in stz induced diabetic rats over 21 days treatment period. Authors also estimated fasting blood glucose levels. The experimental design was simple and easily understood and the experiments seem to be properly and carefully performed. The obtained results and proposed conclusions seem interesting and could help to increase semen quality and fertility in diabetic male individuals. In addition, results might potentially lie the basis for the development of preservative media for mammalian spermatozoa, in which the sperm cells can survive longer and in better shape.

However, before publication the following points should be taken into account

Authors have provided a thoughtful and extensive discussion of the central issues and relevant literature. However, the discussion is much too long, particularly in the proteomic data section (line 488 to 705). This should be significantly reduced by summarizing most of these results. Statistically significant differences should be also shown in the figures. Displaying this information by any means should help to better understand the results.

Round 2

Reviewer 1 Report

Most of the comments were satisfactorily answered. 

However, I suggest further minor text editing.

1) Reference Pavlinkova et al 2017 has reported following: Diabetes decreased sperm concentration and viability and increased sperm apoptosis. Changes in protamine 1/protamine 2 ratio indicated reduced sperm quality. Please, the authors should add this finding to the text of Introduction (l. 54) and part of Discussion (4.2. Sperm Quality Analysis)

2) Response 7 should be added into the text of manuscript, probably in the Discussion part.

3) Response 10: I still lack the exact title of the chapter 3.1 (4.1) Anti-hyperglycemic Activity, but it is not clear "what ?" - sperm? FC ?

4) Discussion looks well, thanks for authors´ improvement. However, in chapter 4.1, the authors should discuss own results with other studies.

5) Could the authors explain what the impact of diabetes (resp. FC treatment) on proteins (Izumo1) involved in the sperm-oocyte interaction? 

6) Response 12 is not satisfactory. Authors should explain protein concentration of sperm sample per lane subjected to SDS-PAGE. Was the sample amount the same in all lanes? Have you used any reference protein to compare the protein concentration in all samples?

Author Response

Thanks a lot for the comments, they are really helped us to improvise the manuscript.

For the details of our reply, please see the attachment.

Thank you.
